# Assessment of Crosslinkers between Peptide Antigen and Carrier Protein for Fusion Peptide-Directed Vaccines against HIV-1

**DOI:** 10.3390/vaccines10111916

**Published:** 2022-11-12

**Authors:** Li Ou, Krishana Gulla, Andrea Biju, Daniel W. Biner, Tatsiana Bylund, Anita Changela, Steven J. Chen, Cheng-Yan Zheng, Nicole Cibelli, Angela R. Corrigan, Hongying Duan, Christopher A. Gonelli, Wing-Pui Kong, Cheng Cheng, Sijy O’Dell, Edward K. Sarfo, Andrew Shaddeau, Shuishu Wang, Alison Vinitsky, Yanhong Yang, Baoshan Zhang, Yaqiu Zhang, Richard A. Koup, Nicole A. Doria-Rose, Jason G. Gall, John R. Mascola, Peter D. Kwong

**Affiliations:** Vaccine Research Center, National Institute of Allergy and Infectious Diseases, National Institutes of Health, Bethesda, MD 20892, USA

**Keywords:** conjugate vaccine, crosslinker, fusion peptide, HIV

## Abstract

Conjugate-vaccine immunogens require three components: a carrier protein, an antigen, and a crosslinker, capable of coupling antigen to carrier protein, while preserving both T-cell responses from carrier protein and B-cell responses from antigen. We previously showed that the N-terminal eight residues of the HIV-1 fusion peptide (FP8) as an antigen could prime for broad cross-clade neutralizing responses, that recombinant heavy chain of tetanus toxin (rTTHC) as a carrier protein provided optimal responses, and that choice of crosslinker could impact both antigenicity and immunogenicity. Here, we delve more deeply into the impact of varying the linker between FP8 and rTTHC. In specific, we assessed the physical properties, the antigenicity, and the immunogenicity of conjugates for crosslinkers ranging in spacer-arm length from 1.5 to 95.2 Å, with varying hydrophobicity and crosslinking-functional groups. Conjugates coupled with different degrees of multimerization and peptide-to-rTTHC stoichiometry, but all were well recognized by HIV-fusion-peptide-directed antibodies VRC34.01, VRC34.05, PGT151, and ACS202 except for the conjugate with the longest linker (24-PEGylated SMCC; SM(PEG)24), which had lower affinity for ACS202, as did the conjugate with the shortest linker (succinimidyl iodoacetate; SIA), which also had the lowest peptide-to-rTTHC stoichiometry. Murine immunizations testing seven FP8-rTTHC conjugates elicited fusion-peptide-directed antibody responses, with SIA- and SM(PEG)24-linked conjugates eliciting lower responses than the other five conjugates. After boosting with prefusion-closed envelope trimers from strains BG505 clade A and consensus clade C, trimer-directed antibody-binding responses were lower for the SIA-linked conjugate; elicited neutralizing responses were similar, however, though statistically lower for the SM(PEG)24-linked conjugate, when tested against a strain especially sensitive to fusion-peptide-directed responses. Overall, correlation analyses revealed the immunogenicity of FP8-rTTHC conjugates to be negatively impacted by hydrophilicity and extremes of length or low peptide-carrier stoichiometry, but robust to other linker parameters, with several commonly used crosslinkers yielding statistically indistinguishable serological results.

## 1. Introduction

An effective antibody-based HIV-1 vaccine, capable of eliciting high titer protective responses, has been a long-sought goal [1,2]. Immunization with prefusion-closed envelope (Env) trimers has enabled the elicitation of autologous-neutralizing responses [3], but broad cross-clade neutralization has been difficult to elicit.

One intriguing preclinical finding is the induction of cross-clade-neutralizing antibodies against the fusion peptide site of vulnerability, with cross-clade neutralizing responses elicited in mice, guinea pigs, and non-human primates [4,5], along with vaccine-elicited antibodies, such as the fusion-peptide-directed DFPH-a.01 (named for NHP-lineage.clone), capable of neutralization 59% of a cross-clade 208-isolate panel [6]. Fusion-peptide-directed immunizations involve two steps: a priming step—focusing immune response to the exposed N-terminal half of the fusion peptide, and a boosting step—involving repeated immunizations with prefusion-closed Env trimers [4,5,7].

The priming step has been successful when placing the N-terminal eight residues of the HIV-fusion peptide (FP8) into multivalent contexts, as exemplified by RNA bacteriophage virus-like particles (VLPs) from MS2 or Qβ VLPs displaying chemically conjugated FP8 peptide [8] or by multivalent scaffolds conjugate immunogens, coupling the most prevalent FP8 peptide (FP8v1) to a highly immunogenic carrier [5]. Conjugate vaccines encompass some of the most successful vaccine modalities against bacterial antigens [9,10] and involve three components: a carrier protein, able to stimulate strong T cell responses [11,12]; an antigen, able to elicit protective-antibody responses [13]; and a crosslinker, capable of linking carrier and antigen, while preserving both T-cell and antibody responses [14,15]. We previously showed FP8 when coupled to the recombinant heavy chain from tetanus toxoid (rTTHC) to be a good priming reagent, although choice of linkers did have an impact [16]. Here, we evaluate additional linkers for coupling fusion peptide to rTTHC as priming immunogens. We created FP8v1-rTTHC conjugates of each and characterized multimerization, stoichiometry, and antigenicity. We immunized mice with linker conjugates, boosted with Env trimers, and measured immune responses. The results provide insight into the impact of linker parameters on fusion-peptide-directed immune response, enabling appropriate choice of linker for clinical assessment.

## 2. Materials and Methods

### 2.1. Ethics Statement

Female 6- to 8-week-old C57BL/6 mice were obtained from Jackson Laboratory (Wilmington, MA, USA). All mice experiments were reviewed and approved in protocol VRC-19-841 by the Animal Care and Use Committee of the Vaccine Research Center, National Institutes of Allergy and Infectious Diseases (NIAID), National Institutes of Health (NIH). All animals were housed and cared for in accordance with American Association for Accreditation of Laboratory Animal Care standards in accredited facilities.

### 2.2. Animal Protocols and Immunization

For each immunization, 25 μg of immunogen formulated with 20% of Adjuplex^®^ (Advanced BioAdjuvants, LLC (ABA), Omaha, NE, USA) in the final volume of 100 μL. Immunizations were performed intramuscularly to the caudle thigh of the two hind legs. Serum was collected 2 weeks after each immunization for serological analysis.

### 2.3. Cell Lines

Expi293F cells were obtained from ThermoFisher Scientific Inc. (Invitrogen, cat# A14528; RRID: CVCL_D615). TZM-bl cells were from NIH AIDS Reagent Program (www.aidsreagent.org (accessed on 1 October 2022), cat# 8129).

### 2.4. Fusion Peptide Immunogens

rTTHC was produced by the VRC Production Program, and HIV-1 fusion peptide (FP8v1: AVGIGAVFC) was synthesized by GenScript (Piscataway, NJ, USA) with a free amine at the N terminus and an extra cysteine residue at the C terminus. rTTHC conjugates were prepared as described previously [16]. Briefly, rTTHC was activated by reaction of its lysine side chains with a bifunctional crosslinker, which was then coupled to C-terminal thiol of FP8v1. Four fusion-peptide-specific antibodies (VRC34.01, VRC34.05, PGT151 and ACS202) were used to characterize the antigenicity of the conjugates. Amino acid analysis was used to define the conjugation ratio of FP to carrier protein.

### 2.5. HIV-1 Envelope Trimers

BG505 DS-SOSIP [17] and a prefusion-stabilized consensus clade C trimer (ConC) with the second most prevalent FP8 sequence (FP8v2) were produced from a CHO-DG44 stable cell line and purified using a series of non-affinity chromatography steps (VRC Production Program) [17].

### 2.6. Antigenic Characterization

The antigenicity of rTTHC conjugates was measured by biolayer interferometry assay (FortéBio Octet HTX) at 30 °C. Samples were diluted in phosphate-buffered saline (PBS) with 1% bovine serum albumin (BSA) to minimize nonspecific interactions. The antibody at 40 ug/mL was immobilized on AHC sensor tips (FortéBio) for 300 s. After equilibrated in buffer for 60 s, the biosensor tips were dipped into two-fold serial dilutions of conjugates for 300 s, followed by 300 s dissociation. Experimental data were analyzed using Global fitting with 1:1 model binding using Octet software, version 9.0.

### 2.7. Anti-Trimer (BG505 DS-SOSIP.664) Enzyme-Linked Immunosorbent Assay (ELISA)

Anti-trimer responses in the immunized mice were analyzed using an in-house developed lectin capture ELISA. The ELISA methodology has been described previously [4]. Briefly, 96-well plates were coated with snowdrop lectin to capture the glycosylated trimer. Serially diluted mouse sera were added and incubated for 1 h at room temperature followed by the goat anti-mouse antibody incubation. Plates were read at 450 nm after developed with tetramethylbenzidine (TMB) substrate for 10 min before the reaction was stopped with 1 N sulfuric acid. Optical densities (OD) were analyzed following subtraction of the nonspecific horseradish peroxidase background activity. Endpoint titer was defined as the reciprocal of the greatest dilution with an OD value above 0.1 (two times the average raw-plate background).

### 2.8. Sera Antigenic Analysis

Fusion-peptide-directed sera responses from rTTHC conjugates immunized mice were assessed by binding to scaffold protein FP8-1M6T on FortéBio Octet HTX. Mouse sera were diluted 1:100 in 1% BSA/PBS, and the naive prebleed sera were used as a reference. The NTA biosensors were used to capture FP8v1-1M6T at 20 μg/mL in 1% BSA/PBS for 300 s followed by equilibration in the buffer for 60 s. The sera responses were measured by the association step for 300 s in sera followed by a dissociation step for additional 60 s. All experiments were performed in duplicate, and the data were analyzed with Octet and GraphPad Prism 8 software.

### 2.9. Flow Cytometry and B Cell Staining

Mouse spleen samples were collected at week 24 and cryopreserved in liquid nitrogen. Single cell suspension of thawed splenocytes was stained sequentially with ViVid and a staining mix containing anti-CD3 Cy55PerCP, anti-CD4 Cy55PerCP, anti-CD8 Cy55PerCP, anti-F4/80 Cy55PerCP, anti-B220 TrPE, anti-IgD BV711, anti-IgM Cy7PE, anti-IgG FITC, and BG505 SOSIP-APC. After 20 min, FP9-PEG12-PE was added and incubated for 20 more minutes. FP9+BG505+ B cells were defined as ViVid−, CD3−, CD4−, CD8−, F4/80−, IgM−, IgD−, IgG+, and FP9+BG505+ B cells. To make antigen-specific probes, biotinylated Avi-tagged FP9-Peg12 and BG505 SOSIP trimer were coupled to Streptavidin-PE and Streptavidin-APC (Life Technologies), respectively.

### 2.10. Neutralization Assays

Neutralization was measured using single-round-of-infection HIV-1 Env-pseudoviruses and TZM-bl target cells, as described previously [4]. We used the Δ611 mutant of BG505 to assess FP-directed responses. This mutant is especially sensitive to FP-directed neutralization. Neutralization curves were fit by nonlinear regression using a 5-parameter hill slope equation. For sera, the 50% and 80% inhibitory dilutions (ID50 and ID80) were reported as the reciprocal of the dilutions required to inhibit infection by 50% and 80%, respectively. Single-point assays were performed in duplicate at a dilution of 50, and data reported as % neutralization.

### 2.11. Statistical Analyses

The statistical difference of neutralization titers or breadth, ELISA titers or B cell frequency between different groups was determined by performing un-paired non-parametric two-tailed Mann–Whitney tests using the GraphPad Prism version9. For correlational analysis between linker properties, serological outputs, and neutralization titers, Spearman’s rank correlation measure was performed using the SciPy spearmanr module (default settings, uncorrected for multiple comparisons). Prior to rank correlation quantification, measurements were averaged over all participants within a linker group. For fair comparison, the geometric mean was the measure of central tendency used for all participant-based measurement averages. To account for weeks with negative neutralization values (for geometric mean quantification), a constant was added to all neutralization measurements (for the week of interest), shifting all values to one or above. For comparison, Spearman’s rank correlation measure was also reported sans the linker length outlier here, SM(PEG)24. Additionally, the linear relationship between correlation pairs was modeled, highlighting relationships which may not reach statistical significance, but fit a simple linear model with relatively low error (R^2^ ≥ 0.8).

## 3. Results

### 3.1. Physical and Antigenic Properties of FP8v1-rTTHC Conjugates with Eight Linkers with Diverse Chemical Properties

Many bifunctional crosslinkers have been used in conjugation of peptide-carrier-based vaccines [15,16,18]. These crosslinkers play crucial roles not only linking vaccine components but also influencing physical and antigenic properties of the conjugate product. We assessed eight crosslinkers with spacer-arm lengths ranging from 1.5 to 95.2 Å and suitable amine- and sulfhydryl-reactive functional groups for conjugating FP8v1 to carrier proteins (Figure 1). The crosslinkers had an iodoacetyl or maleimide group on one end of the spacer arm to link to the reactive sulfhydryl of the Cys appended to the C terminus of FP8v1 and an NHS ester or Sulfo-NHS ester on the other end to react with Lys residues on the surface of rTTHC. All of the crosslinkers were hydrophobic except the PEGylated SMCC crosslinkers [19]. Among them, PEG linkers [20], N-γ-maleimidobutyryl-oxysulfosuccinimide ester (Sulfo-GMBS) [21,22], sulfosuccinimidyl 4-(N-maleimido methyl)cyclohexane-1-carboxylate (Sulfo-SMCC) [23,24] and sulfosuccinimidyl (4-iodoacetyl)aminobenzoate (Sulfo-SIAB) have been used in the clinical products.

All eight crosslinkers, including succinimidyl iodoacetate (SIA) [25,26], N-β-maleimidopropyl-oxysuccinimide ester (BMPS) [27,28], GMBS [29,30,31], SMCC [29,32,33], N-ε-malemidocaproyl-oxysuccinimide ester (EMCS) [34,35,36], SIAB [37,38,39], 2-PEGylated SMCC (SM(PEG)2) [40,41], and 24-PEGylated SMCC (SM(PEG)24) [42,43,44], successfully crosslinked FP8v1 to the recombinant heavy chain of tetanus toxin (rTTHC), with the conjugate products running on SDS-PAGE with molecular sizes larger than rTTHC and with various degrees of multimerization and stoichiometry (Figure 2a,b). The stoichiometry of FP8 conjugated to each rTTHC was around 10 for most of the conjugates, though lower (6.7) for the Sulfo-SIAB-crosslinked product and lowest (3.1) for the SIA-crosslinked product [45]. The Sulfo-SIAB-crosslinked product exhibited the most multimerization compared to the other crosslinkers, though size exclusion chromatography revealed a diverse range of sizes for the conjugated products (Figure 2b).

Next, we assessed the antigenicity of various linker-conjugated FP8v1-rTTHC immunogens for binding to fusion-peptide-directed antibodies elicited by infection VRC34.01 [46], VRC34.05 [46], PGT151 [47], and ACS202 [48] by biolayer interferometry (BLI) (Figure 2c). All of these antibodies bound tightly with apparent binding dissociation constant (K_D_) lower than 1 pM, except for ACS202 binding to the SIA-linked FP8v1-rTTHC with a K_D_ of 0.439 nM (Figure 2c). The generally low apparent K_D_ values resulted from extremely low dissociation rates (k_off_) (Appendix A), reflecting increased avidity of the bivalent immunoglobulin (IgG) binding to multiple FP8v1 on each molecule of rTTHC. The binding on-rates (k_on_) varied for different linkers and antibodies. VRC34.01 exhibited the highest k_on_ of 9.46 × 10^5^ M^−1^s^−1^ for the Sulfo-GMBS-linked conjugate and the lowest k_on_ of 1.08 × 10^5^ M^−1^s^−1^ for the SM(PEG)24-linked conjugate. VRC34.05 also had the lowest k_on_ (2.5 × 10^4^ M^−1^s^−1^) for SM(PEG)24 but had similarly low k_on_ for most linkers except for SIA (k_on_ of 5.7 × 10^5^ M^−1^s^−1^) and SM(PEG)2 (k_on_ of 7.74 × 10^5^ M^−1^s^−1^). PGT151, however, exhibited the highest k_on_ of 2.1 × 10^6^ M^−1^s^−1^ for SM(PEG)24 and the lowest k_on_ of 3.3 × 10^4^ M^−1^s^−1^ for Sulfo-SIAB. By contrast, ACS202 did not bind SM(PEG)24-linked FP8v1-rTTHC and exhibited a measurable k_off_ for the SIA-linked conjugate and a high k_on_ for N-[maleimidocaproyloxy]sulfo succinimide ester (Sulfo-EMCS) [49] and SM(PEG)2 linked conjugates. The binding kinetics of these naturally elicited fusion-peptide-directed antibodies appeared not to correlate with any specific property of the crosslinkers, except with extremes of length or reduced stoichiometry reducing antigenic signatures (Figure 2c and Appendix A). Overall, antigenicity assessments indicated that crosslinkers were able to present FP8v1 on rTTHC, allowing recognition by naturally elicited fusion-peptide-directed antibodies.

### 3.2. Immunogenicity Assessments Indicate FP8v1-rTTHC Conjugates to Be Robust to Most Linker Properties

To assess the impact of the crosslinkers on the immunogenicity of the conjugate products, we carried out immunization studies in mice with seven of the crosslinkers (we did not include a group with SM(PEG)2-linked FP8v1-rTTHC prime because we have previously shown SM(PEG)2 to be inferior to Sulfo-SIAB as a linker for FP8v1-rTTHC immunogen [16]). Seven groups of C57BL/6 mice (n = 10 per group) were immunized three times with FP8v1-rTTHC conjugates (weeks 0, 2, and 4) and boosted twice with BG505 DS-SOSIP [17,50] (weeks 6 and 8) (Figure 3a). A ConC trimer with FP8v2 [7,17,51] was used by itself at week 10 and combined as a cocktail with BG505 DS-SOSIP and used to further boost responses at weeks 12 and 20 (Figure 3a).

We measured serum antibody responses by BLI for binding to fusion peptide (Figure 3b). At week 4, after two immunizations of FP8v1-rTTHC, there were substantial anti-fusion peptide responses with all groups except group 1, which utilized SIA, with the shortest spacer arm of 1.5 Å and the lowest stoichiometry. The fusion-peptide-directed responses at both weeks 4 and 6 were lowest for this group. Titers were also lower for Group 7, which utilized the longest linker for FP8v1-rTTHC at 95.2 Å. Groups 4–6, which had Sulfo-SMCC, Sulfo-EMCS, and Sulfo-SIAB as crosslinkers, exhibited higher anti-FP responses than other groups at weeks 4 and 6. At week 24, after trimer boosts, anti-FP responses substantially decreased for all groups, except group 1 with the SIA linker, with titers generally similar, except for group 7, which remained significantly lower.

Anti-trimer responses were measured by ELISA for binding to BG505 DS-SOSIP (Figure 3c). At week 6, after three FP8v1-rTTHC primes and before any trimer boost, substantial anti-trimer responses were observed for all groups. Similar to the anti-FP response, groups 5 and 6 exhibited higher anti-trimer responses, but no statically significant difference was observed among groups 1–6. Groups 3–6 showed significantly higher anti-trimer response than group 7. At week 8, after the first boost of BG505 DS-SOSIP immunization, ELISA titers increased ~10-fold over week 6 levels for all groups, with all groups showing similar level of anti-trimer response with slightly higher ELISA titers for group 6 but not statistically significant. Further increases in ELISA titers were observed for all groups at week 10, after two boosts of BG505 DS-SOSIP, with group 5 having the highest titers. No further increase in ELISA titers was observed at week 24, with group 1 showing significantly lower titers than other groups.

In addition to fusion-peptide-directed and trimer-directed responses, the number of fusion peptide and trimer double-positive B cells at early stages before extensive priming have been found to correlate [52] and the fraction of anti-base response to inversely correlate with improved neutralizing responses [53]. We therefore assessed the number of double-positive B cells at week 10. The only statically significant differences were observed in groups 3, 4 and 5. These three groups exhibited higher double-positive B cells than group 1 (Figure 3e and Appendix A). To examine if the crosslinkers for the FP8v1-rTTHC conjugates can affect the anti-base response, we measured the amount of anti-base response for week 12 sera by competition ELISA with a base-specific antibody (Figure 3d and Appendix A). Group 1 exhibited the highest anti-trimer base response, while groups 5 and 6 exhibited the lowest anti-base response.

We next analyzed chemical parameters of crosslinkers with the three measured serological results: fusion-peptide-directed responses, trimer-directed responses, and base responses, the first two that correlate positively and the third that correlated negatively with cross-clade neutralizing responses [4,52,53]. Collectively, the results suggested extremes of length, lower stoichiometry or hydrophilicity to trend with reduced immunogenicity of conjugates (Appendix A).

### 3.3. Neutralization Assessment Reveals FP8v1-rTTHC with Hydrophobic Linker of 5–10 A to Be Preferred

Neutralization assessments revealed that at week 10, after three FP primes and two BG505 trimer boosts, none of the groups showed detectable neutralizing activity against BG505 wildtype strain, but substantial neutralizing activity was observed against a variant of BG505 with the N-linked glycan at N611 removed (BG505 Δ611), which is highly sensitive to neutralization by fusion-peptide-directed antibodies [5,46] (Figure 4a,b and Appendix A). At week 24, almost all immunized animals could neutralize BG505 Δ611 and sporadic neutralization was observed against wildtype BG505, though not against the SIV control (Figure 4a–c). Neutralization activity against BG505 Δ611 were slightly weaker for group 7, but no significant difference was observed among the rest of the groups.

We analyzed correlations between neutralizing activity and chemical parameters of the immunogen linker as well as other serological outputs including fusion-peptide- and trimer-directed responses, the number of double-positive B cells at week 10, and the trimer-base responses at week 12 (Appendix A). In terms of BG505 wildtype, we observed several significant Spearman correlations between neutralization activity versus linker and serological properties. Of these, the trimer-base response at week 12 versus neutralization at week 24 showed the highest significance (r(5) = 0.93, *p* = 0.0025), followed by neutralization activity at week 24 versus spacer arm length and linker molecular weight (r(5) = −0.89, *p* = 0.0068 and r(5) = −0.82, *p* = 0.0234, respectively) and between neutralization activity at week 10 versus linker stoichiometry and linker SH reactive group (r(5) = −0.76, *p* = 0.0489 and r(5) = −0.79, *p* = 0.0343, respectively). In terms of neutralization sensitive BG505 D611, the strongest correlations were observed between neutralization activity versus trimer-directed and fusion-peptide-directed responses (e.g., neut. week 24 versus trimer response week 10; r(5) = 0.96, *p* = 0.0005). Correlations of less significance were also observed between neutralization activity versus immunogen antigenicity (e.g., neut. week 10 versus the antigenicity score; r(5) = 0.8, *p* = 0.0301). Overall, although immunogen, linker properties appear to influence neutralization activity against BG505 wildtype at both early and later immunization timepoints, the trimer base response at week 12 emerged as the strongest predictor of neutralization activity here.

Although BG505 neutralization was relatively weak, the data were consistent with the serum antibody binding assays described above in that groups 3, 4, 5 and 6, with spacer arm from 6.8 Å (Sulfo-GMBS) to 10.6 Å (Sulfo-SIAB) trended better overall in eliciting immune response. The results suggest a hydrophobic linker of 5–10 Å coupled with a conjugation stoichiometry of 5–10 fusion peptides linked to each rTTHC to be reasonable linker choices for the FP8v1-rTTHC conjugate as priming immunogen to induce fusion-peptide-directed neutralizing responses.

## 4. Discussion

Recognition of FP8v1-rTTHC immunogens requires antibody to bind the N-terminal residues of fusion peptide, coupled through an appended C-terminal cysteine to a linker, which is in turn coupled to lysine residues on the surface of rTTHC. Here, we assess the impact on antigenicity and immunogenicity of presenting the fusion peptide N terminus with crosslinkers of diverse chemistry, linker length, differing stoichiometries, and differing degrees of rTTHC multimerization. Despite the diverse properties of tested linkers, the overall antigenicity of the linked conjugates was similar. This was also true for the immunogenicity of the elicited neutralizing response, although extremes of linker length, low hydrophobicity, and/or reduced peptide-to-rTTHC stoichiometry may negatively impact the neutralizing response. With only seven groups, we could not fully distinguish between some of these factors, as SIA had both the shortest linker, but also the lowest stoichiometry, and SM(PEG)24 was much longer than the rest of the crosslinkers. Nonetheless, the results indicate that several commonly used linkers including Sulfo-GMBS, Sulfo-SMCC and Sulfo-SIAB elicit statistically indistinguishable neutralizing responses, suggesting them to be suitable choices as linkers for FP8v1-rTTHC vaccines. We note in this context that both Sulfo-GMBS and Sulfo-SMCC have been used clinically [54,55,56] and that GMP grade of Sulfo-SIAB is available for manufacture.

Our results are consistent with studies with a *Salmonella typhimurium* conjugate vaccine, in which the bactericidal activity of serum antibodies induced is independent of the length of the crosslinkers [57], although other studies with Meningococcal capsular polysaccharide vaccines have observed that the nature and length of linkers can impact conjugate immunogenicity [58,59,60]. In several studies, immune responses were observed to linkers themselves [14,61], which we did not observe. These differing outcomes suggest that the impact of linkers can vary with different conjugate-immunogen systems and that each system will likely need experimentally investigation to determine optimal parameters.

Last, we note that the observed neutralization elicited in this study was generally low, especially against wildtype BG505. Moreover, immunizations were only carried out in mice. Before decisions are made for clinical development, it seems wise to assess FP8v1-rTTHC conjugates in additional test species, such as guinea pigs and non-human primates. Overall, however, the results suggest requirements for a hydrophobic linker, but show tolerance for other properties such as linker length (5–10 Å), stoichiometry (5–10 fusion peptides per rTTHC), and reactive chemistry.

## 5. Conclusions

Crosslinkers, coupling antigen and carrier protein, are critical to conjugate vaccines. In this study, we examined the impact of altering the linker between FP8 (the antigen) and rTTHC (the carrier protein). We found that the immunogenicity of the FP8-rTTHC conjugates appeared to be negatively impacted by hydrophilicity and extremes of length or lower stoichiometry. Importantly, however, conjugate immunogenicity was robust to other linker parameters such as degree of multimerization, SH- and NH2-reactive groups, and the molecular weights of linkers. Indeed, several commonly used crosslinkers yielded statistically indistinguishable serological results; thus, some diversity in the choice and chemistry of crosslinker appears to be allowed with FP8-rTTHC conjugates for fusion-peptide-directed vaccines against HIV-1.

## Figures and Tables

**Figure 1 vaccines-10-01916-f001:**
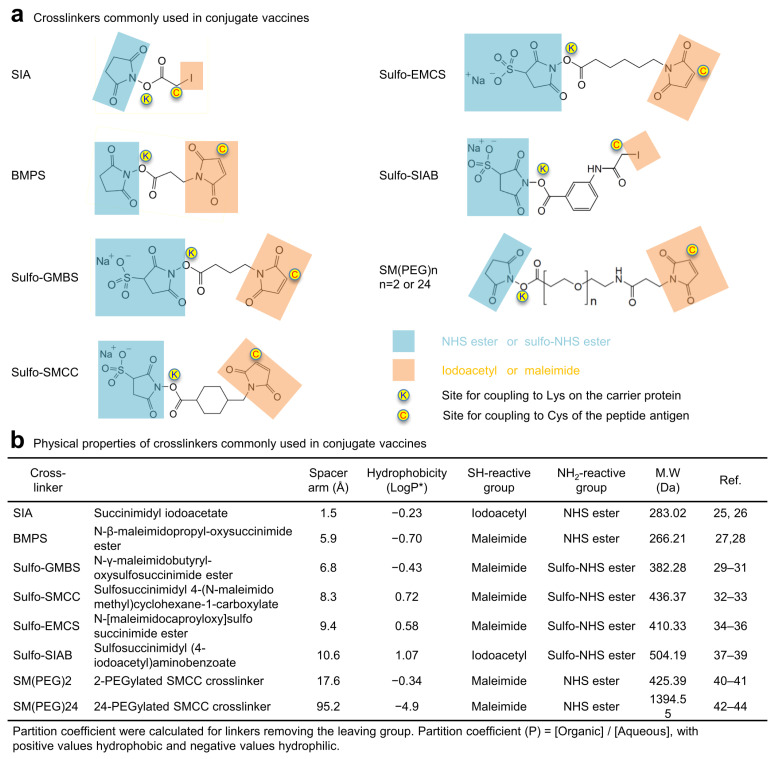
Chemistry of crosslinkers used to conjugate FP8v1 to carrier proteins. (**a**) Structures of crosslinkers used for the conjugation. Functional groups reactive to amino and sulfhydryl groups are highlighted in light blue and orange, respectively. (**b**) Physical properties of crosslinkers commonly used in biomolecule conjugates.

**Figure 2 vaccines-10-01916-f002:**
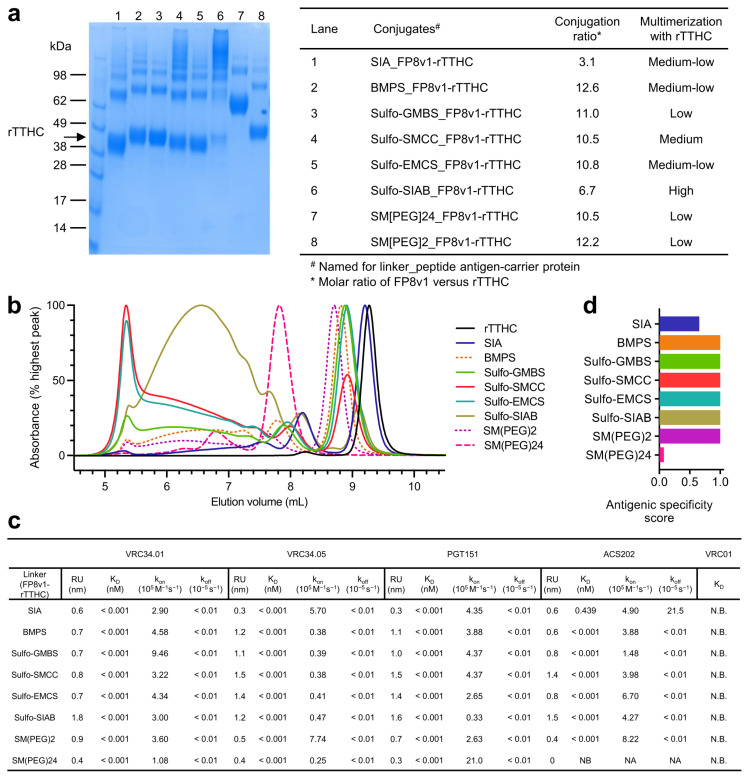
Physical and antigenic properties of FP8v1-rTTHC conjugates with linkers of various spacer-arm length. (**a**) SDS-PAGE of the purified FP8v1-rTTHC conjugates of eight different linkers revealing variable degrees of multimerization. (**b**) Analytical SEC of FP8v1 peptide conjugates. (**c**) Antigenicity characterization of FP8v1-rTTHC conjugates of eight different linkers by BLI measurements of their binding affinity to FP-directed antibodies. (**d)** Antigenic specific score as defined in [5].

**Figure 3 vaccines-10-01916-f003:**
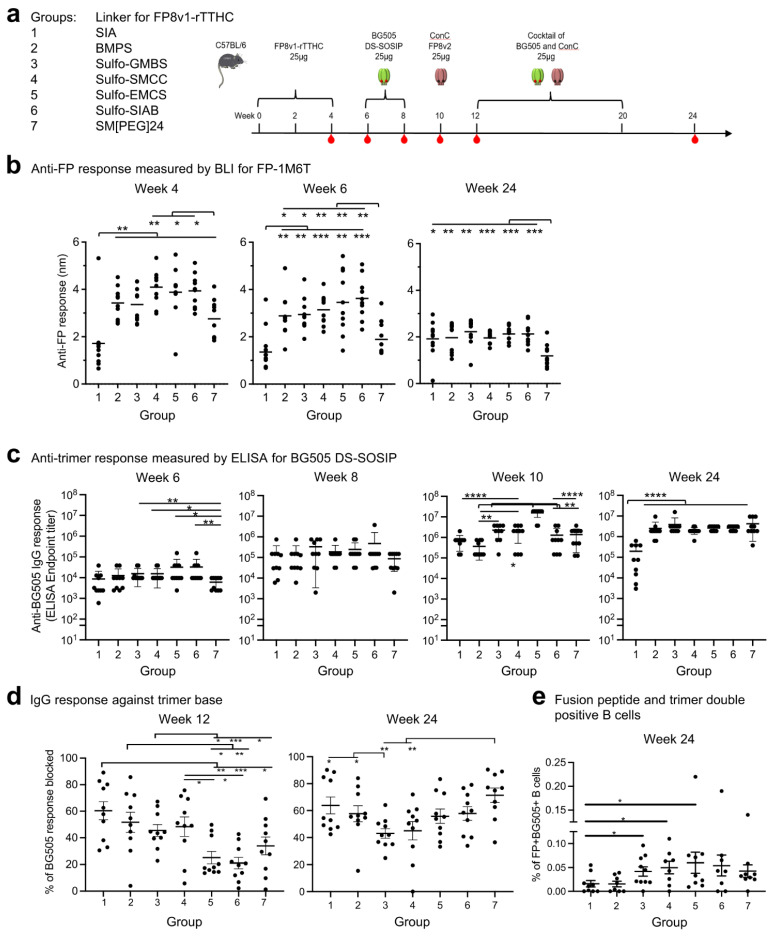
Immunogenicity assessments indicate linker for FP8v1-rTTHC influences antibody responses. (**a**) Immunization regimen. Mice were immunized three times with an FP8v1-rTTHC immunogens and boosted with trimers using Adjuplex as adjuvant. Serum samples were taken at the indicated time points. (**b**) Anti-FP response measured by BLI responses to FP8v1 in the context of the 1M6T-scaffold (PDB: 1M6T). (**c**) Anti-BG505 trimer response. (**d**) Percentages of trimer base responses within total trimer response at week 12 measured by anti-base Fab19R competition assay. (**e**) Fusion peptide and trimer double-positive B cells from week. Two-tailed Mann–Whitney nonparametric tests were used for statistical analysis to assess *p* values for mean ± SEM. *: *p* < 0.05; **: *p* < 0.01; ***: *p* < 0.001; ****: *p* < 0.0001; ns, not significant.

**Figure 4 vaccines-10-01916-f004:**
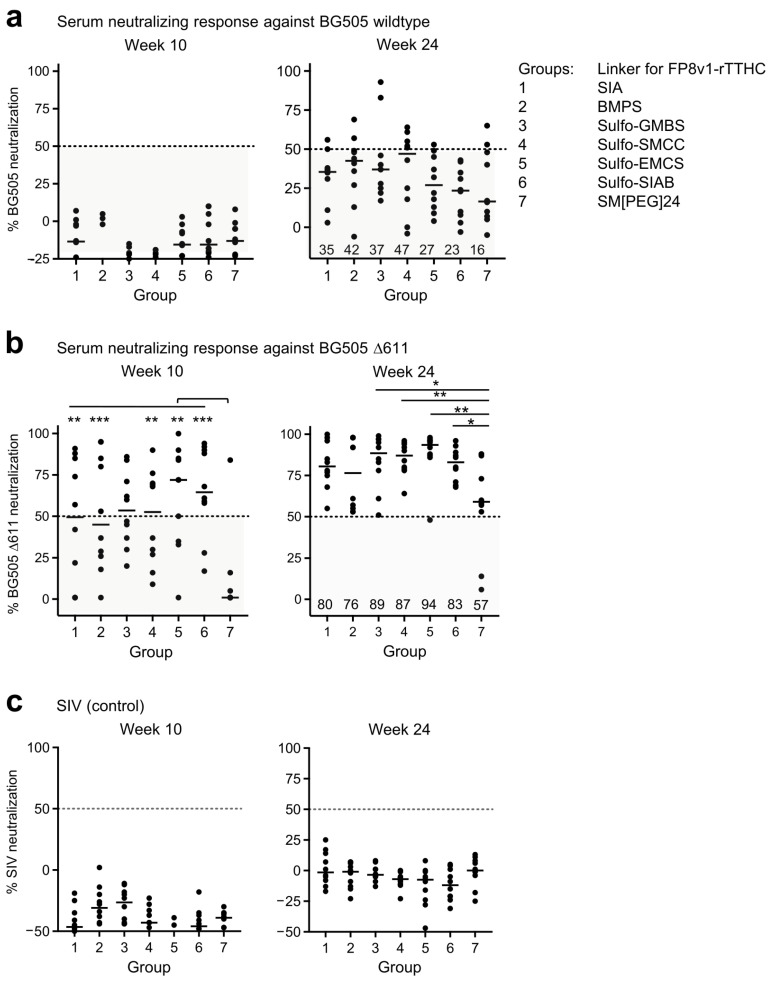
Several linkers for FP8v1-rTTHC yield similar neutralization responses. Virus neutralization assays with the immune sera at week 24 and 10 against (**a**) wildtype BG505, (**b**) BG505 Δ611 virus and (**c**) control SIV. Virus neutralization assays with the immune sera at 1:50 dilution. *p* values were calculated with Mann–Whitney two-tailed t test: * *p* < 0.05; ** *p* < 0.01; *** *p* < 0.001. The values below than 50% are considered to be less potent and shaded in a semi-transparent gray box.

## Data Availability

All relevant data are within the paper and its Appendix A.

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
