# Peer review of "Assessment of Crosslinkers between Peptide Antigen and Carrier Protein for Fusion Peptide-Directed Vaccines against HIV-1"

_vaccines, 2022, doi:10.3390/vaccines10111916_

Round 1
Reviewer 1 Report
This is a straightforward study using appropriate technology and sound conclusions. The limitations of studying exclusively mice are adequately addressed.
This study aimed to determine the relationship between linkers and the immune response to the HIV fusion domain, a conserved region of gp120/41 that might be useful in cross clade immunization. The study used tetanus toxin heavy chain as the 'carrier' protein, they having previously established that it can serve as an appropriate protein to generate an immune response to this small peptide. The investigators found that some linkers were optimal for the generation of a murine antibody response and that the degree of multimerization after the reaction had some influence on the immune response as well, though the directionality was unclear (too short or too long were suboptimal in this limited set). In general, as is stated in the manuscript, one is mostly struck by the relatively low influence of the specific choice of linker in the robustness of the immune response.
The study adds to our understanding of the parameters that might be necessary in order to generate a vaccine against this region of the glycoprotein(s). An effective neutralizing immune response against it would cover many HIV-1 clades. The figures and tables are clear, and the authors correctly point out that a murine system renders only an approximation of the likely human response, and that further studies will be necessary before any potential clinical trial.
Overall this is a clear, well-illustrated paper. The linker figures (1A) are particularly illustrative.
Reviewer 2 Report
Comments and Suggestions for Authors
Manuscript Number: vaccines-1985631
The article entitled Cross-linkers between peptide antigen and carrier protein for fusion peptide-directed vaccines against HIV-1 has been reviewed. The article reported the impact of crosslinking on the immunogenicity of HIV-1 fusion peptide-directed antibodies compared to the recombinant heavy chain of tetanus toxin.
Major comments:
The article did not adequately describe the main aim of the study or provide the main findings of the study. The study is an extended study of the previous study, reference no.16. I have two main concerns about this article.
1. First, the title and conclusion did not agree well with the objective of the study. The authors evaluated the multimerization, stoichiometry, and antigenicity of the FP8v1-rTTHC conjugates (crosslinking FP8 v1 and rTTHC as antigenic) in the study.
2. The results of the study described seem not to be novel; however, the discussion is short and seems to miss many statements, it would be suggested to have a more detailed description that included the antigenicity and T cell responses of the conjugates of FP8v1-rTTHC, and then the limitation of this study, and finally the authors did not confirm the novel conclusion in the study.
Minor comments:
1. The title should clearly state the objective of the study and focus on the impact of crosslinking FP8 v1 and rTTHC on the immunogenicity of vaccines directed by fusion peptides against HIV-1.
2. Please rewrite the statements in the abstract; it should begin with a brief but precise statement of the problem or issue, followed by a description of the research method and design, the major findings, and the conclusions reached.
3. The conclusion of this study should be consistent with the Abstract and Conclusion Sections. However, there is no conclusion in the study.
4. Please rewrite the statements including 2.1. Ethics Statement, 2.2. Animal protocols and immunization, 2.3. Cell lines, 2.4. Fusion peptide immunogens, 2.6. Antigenic characterization, and 2.8. Sera antigenic analysis (P2 L69~ P 4 L 152). The description is copied to the original paper, reference No.16.
5. The statement 'Statistical analyses' is lacking.
6. Provide the limitation of this study in the discussion section that could be expected.
Reviewer 3 Report
Due to its inherent variability, development of a universal vaccine against HIV-1 has proven to be extremely difficult and has required an iterative approach. In that vein, the authors, armed with an immunogen (FP8) and a carrier protein (rTTHC), have investigated the efficacy of a series of linker molecules in eliciting antibody responses to FP8 and subsequently neutralizing antibodies to intact HIV-1 trimers. Of the seven linkers tested, most provoked similar results, with perhaps 2 linkers showing a less robust antigenic profile. These two were composed of the longest and shortest linkers and had the lowest stoichiometry of FP8:rTTHC. However, although antibody responses were not statistically different for most of the linkers, none of them were able to elicit a strong neutralizing response against wild type BG505 trimers, although there were stronger responses against a glycan-deleted variant (delta611) of BG505.
As expected from this group, the manuscript is a well written and controlled study, and the results/conclusions are valid. The immunogen design clearly needs more work, as expected for this type of iterative project.
Minor points
1. Early in the results section, the authors should make it clearer than the rTTHC molecule has multiple lysine residues that can crosslink the linker (it is glossed over in the brief discussion section), while the peptide only has a single cysteine for cross-linking and that this is the reason for the stoichiometry observed. If there are any hypotheses why some linkers gave lower ratios, it should be included in the discussion.
2. There seems to be an editing error on line 230. Please clarify.
Round 2
Reviewer 2 Report
Journal
Vaccines (ISSN 2076-393X)
Manuscript ID
vaccines-1985631
Type
Article
Title
Crosslinkers between peptide antigen and carrier protein for fusion peptide-directed vaccines against HIV-1
Major Comments:
The results of the study described seem not to be novel; however, the discussion is short and seems to miss many statements, it would be suggested to have a more detailed description that included the antigenicity and T cell responses of the conjugates of FP8v1-rTTHC, and then the limitation of this study, and finally, the authors did not confirm the novel conclusion in the study.
Response:
The authors stated that “the results describing impact of different crosslinkers on multimerization, stoichiometry, and antigenicity of the FP8v1-rTTHC conjugates are novel. We appreciate this reviewer suggesting that we clarify conclusions, which we have now added as a new last section to the paper.”
However, this did not appear in the revised manuscript. There was rarely a study in the discussion section that had no studies to compare and cited any reference.
